# Multicolor Luminescent Carbon Dots: Tunable Photoluminescence, Excellent Stability, and Their Application in Light-Emitting Diodes

**DOI:** 10.3390/nano12183132

**Published:** 2022-09-09

**Authors:** Longshi Rao, Qing Zhang, Bin Sun, Mingfu Wen, Jiayang Zhang, Guisheng Zhong, Ting Fu, Xiaodong Niu

**Affiliations:** 1Department of Mechanical Engineering, College of Engineering, Shantou University, Shantou 515063, China; 2Intelligent Manufacturing Key Laboratory of Ministry of Education, Shantou University, Shantou 515063, China; 3Hubei Key Laboratory of Mechanical Transmission and Manufacturing Engineering, Wuhan University of Science and Technology, Wuhan 430081, China

**Keywords:** carbon dots, multicolor emission, solvent engineering, light-emitting diodes

## Abstract

Carbon dots (CDs) are attracting much interest due to their excellent photoelectric properties and wide range of potential applications. However, it is still a challenge to regulate their bandgap emissions to achieve full-color CDs with high emissions. Herein, we propose an approach for producing full-color emissive CDs by employing a solvent engineering strategy. By only tuning the volume ratio of water and dimethylformamide (H_2_O/DMF), the photoluminescence (PL) emission wavelengths of the CDs can be changed from 451 to 654 nm. Different fluorescence features of multicolor CDs were systematically investigated. XRD, SEM, TEM, Abs/PL/PLE, XPS, and PL decay lifetime characterizations provided conclusive evidence supporting the extent to which the solvent controlled the dehydration and carbonization processes of the precursors, leading to a variation in their emission color from red to blue. The as-prepared CDs exhibited excellent and stable fluorescence performance even after being heated at 80 °C for 48 h and with UV light continuously irradiated for 15 h. Based on their excellent fluorescent properties and photothermal stability, bright multicolor light-emitting diodes with a high CRI of up to 91 were obtained. We anticipate that these full-color emissive CDs are beneficial for applications in lighting, display, and other fields.

## 1. Introduction

Since the discovery of fluorescence emission from quantum-sized carbon dots (CDs, <10 nm), which was accidentally observed during the purification of a single-walled carbon nanotube solution, light-emitting CDs have triggered intense research interest [1,2,3]. Due to their outstanding advantages of low cost, excellent optical properties, biocompatibility, and environmental friendliness, CDs are considered a high potential candidate to replace heavy metal-based semiconductor quantum dots for light-emitting devices, sensors, biomedical, imaging, energy storage, and other applications [4,5,6,7,8].

To date, hundreds of approaches and thousands of raw materials have been devoted to CDs to achieve better photoluminescence quantum yields (PLQYs), explore their physicochemical properties, and develop their applications. The methodologies mentioned are divided into two main categories: “top-down” and “bottom-up” [9,10,11,12]. Using the “top-down” method, the synthesis of CDs usually involves the destruction of bulky carbon materials such as graphite through arc discharge, laser ablation, and other energy-intensive techniques [13,14]. The “bottom-up” method requires high temperature, high pressure, and strong oxidative reagents to prepare CDs from small molecular precursors such as amino acids and carbohydrates [15]. CDs with a high PLQY can be easily manufactured using the synthesis procedures described above. Despite the great success achieved in PLQYs, most CDs only exhibit fluorescence emission in the blue-to-green-light region [16,17]. However, multicolor, long-wavelength, and white light-emitting CDs are of great importance in applications such as bioimaging, multicolor patterning, white light-emitting diodes (LEDs), sensor arrays, and full-color displays [18], attracting considerable interest in the preparation of CDs with specific optical properties.

The multicolor emission of CDs cannot be obtained by simply adjusting the particle size, as is not the case with semiconducting quantum dots. In fact, in most cases, the multicolor emission of CDs is caused by the surface functional groups rather than by their size [18]. Up to now, various methods, mainly including solvent engineering [19], surface modification [20], precursor introduction [17], heteroatom doping [21], and surface defect regulation [22], have been developed for achieving multicolor and long-wavelength CDs. Among these methods, solvent engineering is an effective way to tune the fluorescent wavelength and enhance the fluorescence intensity of CDs. Previous reports on the influence of solvents in preparing CDs have proved that this factor is critical to achieving multicolor effects [23]. Xiong et al. demonstrated the synthesis of full-color CDs by tuning the ratios of reactive molecules and different solvents [24]. Similarly, Wang et al. showed that CDs emitted different PL colors when in different solvents [25]. Zhan et al. obtained full-color CDs by varying the compositions of reaction solvents [26]. Although several CDs with tunable emissions have been successfully obtained, their complicated synthesis still requires different precursors and solvents, which greatly limits the development of CD mechanisms and applications. In addition, in pursuit of a high-quality light-emitting device, especially LEDs with high color rendering index (CRI, >80), it will undoubtedly be necessary to use practical synthesis techniques to create robust CDs with tunable PL emission. Therefore, the development of energy-efficient, safe, and highly stable synthetic methods for multicolor CD emissions remains to be done.

In this study, we demonstrate a one-step solvent engineering strategy for producing highly emissive CDs with remarkably tunable and stable fluorescence emissions from red to blue. By controlling the volume ratio of water and dimethylformamide (H_2_O/DMF), the PL emission wavelengths of the as-prepared CDs can be tuned from 451 to 654 nm. Various techniques were used to gain a deeper understanding of the engineering mechanism of solvents, including XRD, SEM, TEM, Abs/PL/PLE, XPS, and PL decay lifetime analysis. The results provide solid evidence supporting the extent to which the solvent played a substantial role in the dehydration and carbonization processes of the precursors, leading to variations in their emission colors from red to blue. Furthermore, the as-prepared CDs exhibited excellent fluorescence performance even after being heated at 80 °C for 48 h and with UV light continuously irradiated for 15 h. Taking advantage of their excellent fluorescent properties and photothermal stability, bright multicolor light-emitting diodes with a high CRI of up to 91 were obtained, paving the way to practically applicable LEDs. We anticipate that these full-color emissive CDs will be beneficial for applications in lighting, display, and other fields.

## 2. Materials and Methods

### 2.1. Chemicals and Materials

We obtained aqueous ammonium citrate (A.R.), N, N-dimethylformamide (DMF, 99.8%), and petroleum ether (PE, A.R.) from Aladdin Biochemical Technology Co., Ltd. (Shanghai, China). Polydimethylsiloxane (PDMS) was purchased from Dow Corning (Shanghai, China). The scientific reagents used in this experiment were used as received without further purification. Deionized water (H_2_O, 18.2 MΩ) was used for all experiments.

### 2.2. Synthesis of Multicolor Luminescent CDs

Multicolor luminescent CDs were synthesized using H_2_O and DMF as solvents and ammonium citrate as a precursor. Typically, red-emission CDs (R-CDs) are prepared using 3.9 g of ammonium citrate dissolved in 5 mL of H_2_O and 45 mL of DMF. These solutions were magnetically stirred until well-mixed. A light yellow solution was obtained by transferring the mixture to a Teflon-coated stainless steel autoclave and heating it for 8 h at 200 °C. Then, the products were purified three times with PE to remove impurities, and they were dialyzed (1000 Da of cut-off molecular weight) for 24 h to remove unreacted precursors. Finally, the bright emissive R-CDs solution was achieved. Other yellow-emitting CDs (Y-CDs), green-emitting CDs (G-CDs), and blue-emitting CDs (B-CDs) were obtained by only varying the volume ratio of the H_2_O and DMF in the mixture. Other experimental procedures were carried out as described above, as shown in Figure 1 and Appendix A. In detail, 20 mL of H_2_O and 30 mL of DMF (20:30) were used to prepare the Y-CDs, 30 mL of water and 20 mL of DMF (30:20) were used for the G-CDs, and 50 mL of water (50:0) was used for the Y-CDs.

### 2.3. Fabrication of CD-Based LED Devices

We created CD-based LEDs in a similar way to our previously published work [27]. Using R-CD LEDs as an example, R-CDs were combined with PDMS at a weight ratio of 3:8 for 12 min using a vacuum homogenizer at 1360 rpm and 0.2 MPa vacuum. To form a phosphor layer, the mixture was dropped onto the surface of the UV LED chip and solidified at 120 °C for two hours under ambient conditions. To prevent contamination and damage, a high transmittance hemispherical lens was used for encapsulation.

### 2.4. Characterization

The crystal phases of the products were determined using X-ray diffraction (XRD, D8-ADVANCE, Bruker, Karlsruhe, Germany) with a Cu-Kα radiation source (λ = 0.15418 nm) at 35 kV and a counting rate of 2°/min in the 5° to 60° scanning angle range. A transmission electron microscope (TEM, JEM-2100F, JEOL, Tokyo, Japan) with a 200 kV accelerating voltage was used for the observations. The UV-Vis absorption spectra of the samples were obtained using a UV-Vis spectrometer (Tu-1901, Purkinje, Beijing, China). A fluorescence spectrophotometer (RF-6000, Shimadzu, Kyoto, Japan) with a xenon lamp as an excitation source was used to record the PL spectra of the products. A Fourier transform infrared spectrometer was used to record FTIR spectra ranging from 4000 to 400 cm^−1^ (Vertex 33, Bruker, Karlsruhe, Germany). A Thermo Scientific (Thermo K-Alpha, T.F.S., Waltham, MA, USA) machine with a mono Al-Kα excitation source (1486.6 eV) as the X-ray source was used for X-ray photoelectron spectroscopy (XPS). Edinburgh Instruments were used to collect the PL lifetime (FLS 980, E.I., Edinburgh, UK). The PL decay curves obtained were fitted with the multiple exponential functions given in the expression below [28].
(1)$$A(t)=\sum_{i=1}^{n}A_{i}\exp(\frac{-t}{\tau_{i}})$$
where *A*(*t*) represents the PL intensity at time *t*; *A*_i_ denotes the relative weights of the lifetime components at time *t* = 0; *τ*_i_ represents the decay time for the lifetime components. The average decay lifetime *τ*_avg_. was calculated using the following expression [29]:(2)$$\tau_{avg.}=\frac{\sum_{i=1}^{n}A_{1}\tau_{i}^{2}}{\sum_{i=1}^{n}A_{1}\tau_{i}}$$

## 3. Results and Discussion

As shown in Figure 2, by only changing the solvent volume ratio of the H_2_O/DMF, the PL emission wavelengths of the as-prepared samples could cover the visible spectrum ranging from 451 to 654 nm, which fully demonstrates the feasibility and convenience of the solvent engineering strategy. Furthermore, to investigate the effect of the H_2_O/DMF volume ratio on the optical properties, morphological features, and multicolor light-emitting mechanism of the as-prepared samples, we focused on four typical CDs—R-CDs, Y-CDs, G-CDs, and B-CDs—which were prepared using the H_2_O/DMF volume ratios of 5:45, 20:30, 30:20, and 50:0, indexing as 45D, 30D, 20D, and 0D, respectively, as presented in Appendix A.

### 3.1. Morphological and Structural Characterizations of Multicolor Luminescent CDs

To confirm the nature of the carbon nanoparticles, the as-prepared CDs were characterized using XRD and TEM. The XRD patterns of the R-CDs, Y-CDs, G-CDs, and B-CDs exhibited a typical carbon structure feature with significant diffraction peaks, as shown in Figure 3a,d,g,j. These four CDs demonstrate a single broad diffraction peak at 2θ = 29.4°, 29.6°, 29.6°, and 29.2°, respectively, which corresponds to the (002) *hkl* plane of the graphite carbon structure (JCPDS PDF # 75-2078) and proves the formation of a very tiny carbogenic core in CDs [30,31,32]. According to the Debye–Scherrer equation, the XRD pattern peaks are broadened when the crystal size is decreased:(3)D=kλ/βcosθ
where *β* is the width of the observed diffraction peak at its half-maximum intensity (FWHM), *k* is the shape factor, which takes a value of about 0.9, *θ* is Bragg’s angle, *D* is the crystallite size, and *λ* is the X-ray wavelength (Cu-Kα radiation, equal to 0.15444 nm).

Furthermore, TEM was used to examine the morphology and size of the four CDs. Figure 3b,e,h,k shows that the four CDs were uniform and well-dispersed. Meanwhile, the four CDs’ size distribution histograms (Figure 3c,f,i,l) show a narrow size distribution, with average diameters of 2.36 ± 0.1, 2.27 ± 0.1, 2.18 ± 0.1, and 2.96 ± 0.1 nm, respectively. According to the TEM data, the average diameter of the corresponding CDs decreased and then increased from the R-CDs to the B-CDs, and some large crystals appeared. Notably, when compared to the R-CDs, Y-CDs, and G-CDs, the obtained B-CDs easily formed “clusters” due to their high surface energy and agglomeration tendency, which can explain why the particle sizes of the B-CDs calculated by the Debye–Scherrer equation differ from those using TEM.

### 3.2. Optical Properties of the CDs

The UV-Vis absorption and PL emission spectra were conducted by UV-visible spectroscopy and photoluminescence spectroscopy to evaluate the optimal optical properties of the R-CDs, Y-CDs, G-CDs, and B-CDs, as shown in Figure 4a–f. The R-CDs, Y-CDs, G-CDs, and B-CDs exhibited rose red, dark yellow, yellow, and light yellow when exposed to sunlight, and they emitted bright red, yellow, green, and blue light when exposed to 365 nm ultraviolet radiation (Figure 4a,b). Surprisingly, the UV-Vis absorption and PL emission of the as-prepared CDs could be tuned by changing the volume ratio of H_2_O/DMF while keeping the other conditions constant. Figure 4c–f clearly shows that these CDs’ absorption characteristics differ from UV to visible wavelengths. All four CDs had a strong absorption band in the 200–300 nm range in the high-energy region, corresponding to the C=C bond π-π* transition in the sp^2^ carbon domain [24]. Nevertheless, in the low-energy region, their characteristic absorption peaks differed significantly. The absorption bands of the R-CDs, Y-CDs, G-CDs, and B-CDs were distinct at 338 nm, 328 nm, 324 nm, and 320 nm, respectively, and are attributed to the n–π* transitions of the surface states containing C=N and C=O structures [24]. Because the absorption of the Y-CDs and G-CDs was stronger than that of the R-CDs and B-CDs, the former two samples had more C=C, C=N, and C=O groups on their surface [33]. Furthermore, the similar UV-Vis absorption properties of the R-CDs, Y-CDs, G-CDs, and B-CDs suggest that these four CDs were formed through similar processes. The intensities of each peak, however, differed, depending on the carbon core and surface state of each sample.

The R-CDs, Y-CDs, G-CDs, and B-CDs had PL emission peaks at 645 nm, 586 nm, 530 nm, and 451 nm, respectively, indicating that the PL emission range of these samples covered the entire visible light region. The Stokes shift between the first characteristic absorption peak and the PL emission was more than 130 nm. The greater the Stokes shift, the less overlap there is between the absorption and emission spectra, which can prevent fluorescence efficiency from being reduced due to energy transfer and benefit from achieving a higher quantum yield. Furthermore, we discovered that the PL emission peaks from the R-CDs to B-CDs gradually became symmetrical and smooth, as opposed to unsymmetrical or sharp peaks, implying an increase in the CD purity. The full widths at the half-maximum (FWHM) of the R-CD, Y-CD, G-CD, and B-CD PL emissions were 93.7 nm, 118.8 nm, 136.4 nm, and 73 nm, respectively. The B-CDs and R-CDs with lower FWHM values had higher color purity, which is advantageous for lighting and display applications.

The PL emission spectrum, normalized PL emission spectrum, and three-dimensional excitation–emission fluorescence spectrum of the R-CDs, Y-CDs, G-CDs, and B-CDs under different excitation wavelengths are presented in Figure 5 to demonstrate the relationships between the excitation and emission properties of CDs. As the excitation wavelengths increased, the maximum emission wavelengths of the R-CDs and B-CDs showed no significant changes, indicating relatively stable PL emissions at 650 nm and 452 nm, respectively (Figure 5a–c,j–l). This excitation-independent feature was most likely caused by an eigenstate emission, which is closely related to the carbon core and is more similar to traditional inorganic phosphors than previously reported CDs [34]. However, as the excitation wavelengths increased, the maximum emission wavelengths of the as-prepared Y-CDs and G-CDs gradually redshifted (Figure 5d–i), demonstrating excitation-dependent PL features, similar to most CDs in previous works [35,36]. The excitation-dependent PL behaviors are possibly related to the carbon bonds and surface functional groups [37,38]. These findings are consistent with UV-Vis absorption characteristics.

### 3.3. Structure Analysis and Multicolor Spectral Regulation Mechanism of the CDs

To further explicate the chemical structure of the as-obtained samples, FTIR and XPS were employed to identify the chemical bonding and chemical compositions of the R-CDs, Y-CDs, G-CDs, and B-CDs. These CDs had similar FTIR spectra, as shown in Figure 6a. They all exhibited the clearly distinct absorption peaks of O–H/N–H at approximately 3446 cm^−1^, C=O at 1710 cm^−1^, C=C/C=N at 1650 cm^−1^, C–N= at 1400 cm^−1^, C–O at 1180 cm^−1^, and C–H at 870 cm^−1^ [37,39]. The stretching vibrations of O–H/N–H, in particular, were allocated to the absorption band at 3446 cm^−1^, confirming the creation of –OH during the synthesis of the four CDs. The stretching vibrations of C=O are indicated by the absorption bands located around 1710 cm^−1^. In addition, the weak absorption band centered at 1650 cm^−1^ is attributed to the C=C/C=N characteristic absorption bands. During the synthesis process, the stretching vibrations band of C–N= (1400 cm^−1^) and C–O (1180 cm^−1^) were observed, indicating the formation of polyaromatic structures in the four CDs. The C–H bending vibration was also detected at 870 cm^−1^. Following further investigation, we discovered that the stretching vibration intensities of C=C/C=N (1650 cm^−1^) gradually decreased or shifted to higher wavenumbers from the R-CDs to the B-CDs, whereas the stretching vibration intensities of C=O (1710 cm^−1^), C–N= (1400 cm^−1^), and C–O (1180 cm^−1^) slowly increased, indicating that the content of the nitrogen-doping polyaromatic structures gradually increased and the degree of carbonization decreased [40]. These findings imply that the chemical structures of the CDs had a significant impact on the PL emissions.

The presence of chemical structures in the R-CDs, Y-CDs, G-CDs, and B-CDs was further confirmed by XPS. The full XPS spectra of the four CDs exhibited three characteristic peaks at 284.5 eV, 399.5 eV, and 531.4 eV, corresponding to C 1s, N 1s, and O 1s (Figure 6b), respectively, indicating that nitrogen atoms were doped into the CD framework. The C 1s band can be separated into three binding energy peaks at 284.5 eV, 286.0 eV, and 287.8 eV in the high-resolution XPS spectra (Figure 7a–l), which can be ascribed to the sp^2^ carbon (C=C/C–C), sp^3^ carbon (C–N/C–O), and carbonyl groups (C=O/C=N), respectively. The N 1s spectra show three component peaks at 398.6 eV, 399.4 eV, and 400.4 eV, corresponding to pyridinic N, pyrrolic N, and graphitic N [41]. The component peaks of the O 1s band are 531.1 eV and 532.6 eV, corresponding to C=O and C–O, respectively. The blue shift in the emission wavelength decreased with a decreasing carbon concentration and decreasing degree of carbonization, as determined by carefully comparing the relative strength of the C 1s peak. In addition, the N/C ratios of the R-CDs, Y-CDs, G-CDs, and B-CDs were 0.12, 0.13, and 0.16, respectively, and the O/C ratios were 0.23, 0.26, 0.29, and 0.29, respectively (Figure 7m–p), indicating that the nitrogen and oxygen contents increased progressively from the R-CDs to B-CDs. The rising oxygen concentration may be interpreted as a measure of the reduced degree of dehydration and carbonization of the precursors in solvothermal synthesis [5], which is supported by the FTIR data analysis. In general, the FTIR and XPS data show that the shifted absorption and emission bands were connected to the chemical structure and degree of carbonization.

The time-resolved decay curves were then used to investigate the PL dynamics for the R-CDs, Y-CDs, G-CDs, and B-CDs. The PL decay times of these four CDs were measured using a 375 nm pulse laser as an excitation source. According to Equations (1) and (2), each PL decay curve can be fitted to a double exponential formula, as illustrated in Figure 8a–e and Table 1, indicating that the luminescence processes of these four samples were identical. In particular, all CDs display similar characteristic time constants τ_1_ and τ_2_ in fitting curves, showing that all four samples included several emitting species with varying recombination rates [28]. The quick component τ_1_ corresponds to eigenstate radiative recombination, whereas the slow component τ_2_ relates to surface state recombination processes [33]. By precise calculation, the fitted τ_1_ values for the R-CDs, Y-CDs, G-CDs, and B-CDs were 14.56 ns, 10.76 ns, 1.24 ns, and 1.19 ns, respectively, and the fitted τ_2_ values for the R-CDs, Y-CDs, G-CDs, and B-CDs were 21.38 ns, 10.86 ns, 6.88 ns, and 6.72 ns, respectively. The R-CDs, Y-CDs, G-CDs, and B-CDs had computed average τ_avg_ values of 18.55 ns, 10.81 ns, 6.28 ns, and 6.27 ns, respectively. The existence of τ_1_ and τ_2_ implies that the CDs had two luminescence centers, coming from the π-π* transitions of the carbon core with conjugated sp^2^ domains and the n-π* transitions of the O-based and N-based contained functional groups, which adjusted the fluorescence performance of the CDs jointly. In addition, the average τ_avg_ values suggest that the carbon core influenced the PL decays of the R-CDs and Y-CDs, whereas the surface functional groups of the CDs governed the PL decay of the G-CDs and B-CDs. As a result, these materials’ multicolor fluorescence features were intimately connected to their carbon core and surface state.

By simply varying the volume ratios of the solvents, four CDs with distinct light-emitting wavelengths were created in this study. As shown in Figure 2, the PL emission wavelengths of the CDs ranged from 451 to 654 nm when the volume ratios of H_2_O/DMF fluctuated from 50/0 to 0/50, demonstrating that solvent characteristics resulted in emission wavelength redshift. Previous research has shown that aprotic solvents, such as DMF, are more likely to provide CDs precursors with a higher degree of dehydration and carbonation than protic solvents such as H_2_O, resulting in different sp^2^ carbon domains, and thus, a different spectrum ranging from blue to red regions [5], which is consistent with XPS characterization. In addition, the solvent solubility and boiling point had a considerable impact on the chemical structure and morphology of the CDs, resulting in varied emission wavelengths. As we know, the precursor ammonium citrate is highly soluble in water but less soluble in DMF at room temperature, which can be attributed to DMF’s methyl groups possessing steric barriers that decrease solvent–solvent interaction [42]. As a result, the solubility of the ammonium citrate in the solvent steadily decreased as more DMF was employed, and the emissions of the produced CDs shifted toward a long wavelength. Furthermore, the boiling temperatures of DMF and H_2_O are 153 °C and 100 °C, respectively. The lower boiling point provided higher reaction pressure in the autoclave under the same reaction conditions, which greatly simplified the doping of N and O inside the produced CDs. Thus, the nitrogen and oxygen contents gradually increased from the R-CDs to the B-CDs. Therefore, the polarity, solubility, and boiling point of the solvent all had an impact on the shifts in the CD emissions.

Stability is a crucial factor in determining the dependability of materials. The fluorescence characteristics of the R-CDs, Y-CDs, G-CDs, and B-CDs were studied after heat treatment and long-term UV irradiation to value their thermal stability and photostability, as shown in Figure 9. When the four CDs were heated at 80 °C for varying times, their fluorescence intensities steadily decreased with time, with the R-CDs declining the most. After 48 h of heating, the fluorescence intensities of these four CDs persisted between 86 and 92% (Figure 9a). Furthermore, after 15 h of continuous illumination at room temperature with a 365 nm UV light, the PL intensities of the four CDs remained at 87–93% of their starting values, as shown in Figure 9b. Based on the foregoing results, all four CDs exhibited high PL stability, making them promising for practical applications.

### 3.4. Application in LEDs

Due to their good fluorescence qualities and great thermal and photo-stability, the R-CDs, Y-CDs, G-CDs, and B-CDs were employed to manufacture multicolor light-emitting diodes (LEDs). The excitation source was a 365 nm UV LED chip. The electroluminescence (EL) spectra of the CD-based monochrome LEDs shifted from red to blue by varying the mass ratios and concentrations of these four CDs. Figure 10a–d shows the typical EL spectra of the four CD-based LEDs, with intense red, yellow, green, and blue light and prominent PL peaks at 650 nm, 593 nm, 532 nm, and 448 nm, respectively. The CIE color coordinates of the as-prepared R-CD-, Y-CD-, G-CD-, and B-CD-based LEDs were (0.60, 0.37), (0.46, 0.47), (0.31, 0.42), and (0.20, 0.23), respectively, and the CCT values were 2010, 3132, 6156, and 13,156 K. Furthermore, the CRI of the R-CD-based LED lamp was 91, which was much higher than the CRIs of the Y-CD-, G-CD-, and B-CD-based LED lamps (CRI = 70, 78, and 89, respectively), indicating that the R-CD-based LED lamp provided high-quality light. The multicolor LEDs indicate that the CDs are promising candidates for display and lighting.

## 4. Conclusions

In summary, we developed a method for adjusting the mixed solvent volume ratios in a solvothermal synthesis employing two independent solvents (H_2_O and DMF) and their mixture. By simply changing the volume ratio of the H_2_O/DMF from 50/0 to 0/50, the CDs’ PL emission wavelengths spanned from 451 to 654 nm. All of these XRD, SEM, TEM, Abs/PL/PLE, XPS, and PL decay lifetime characterization results provide solid evidence supporting the extent to which the solvent controls the dehydration and carbonization processes of the precursors, resulting in the as-prepared CDs with different chemical structures and surface states and a change in emission colors from red to blue. They had both strong thermal characteristics and steady fluorescence performance, ranging from the R-CDs to the B-CDs. After 48 h of heating, the fluorescence intensities of these four CDs persisted for between 86 and 92%. Furthermore, the PL intensities of the four CDs were sustained at 87–93% of their original values after 15 h of continuous stimulation. Multicolor CD-based LEDs with the highest CRI of 91 were created using exceptional fluorescence characteristics and photothermal stability. The full-color emissive CDs are expected to be beneficial in applications such as lighting, display, and other fields.

## Figures and Tables

**Figure 1 nanomaterials-12-03132-f001:**
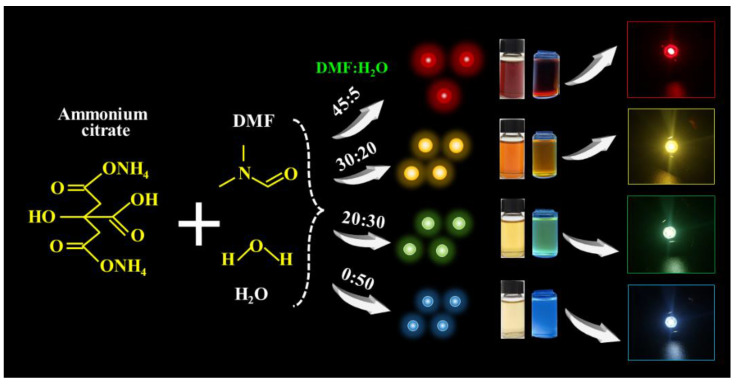
The procedures for the preparation of multicolor CDs and their application in LEDs.

**Figure 2 nanomaterials-12-03132-f002:**
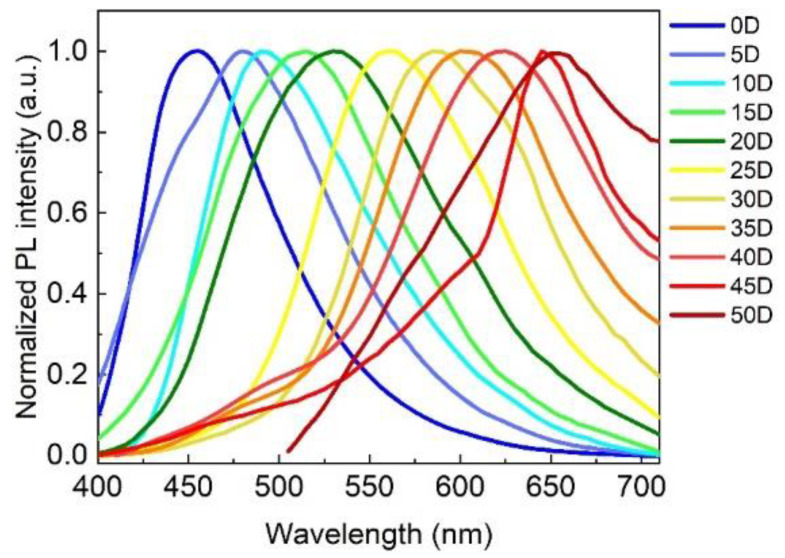
PL spectra of the as-prepared samples that were synthesized by changing the solvent volume ratio between the H_2_O and DMF.

**Figure 3 nanomaterials-12-03132-f003:**
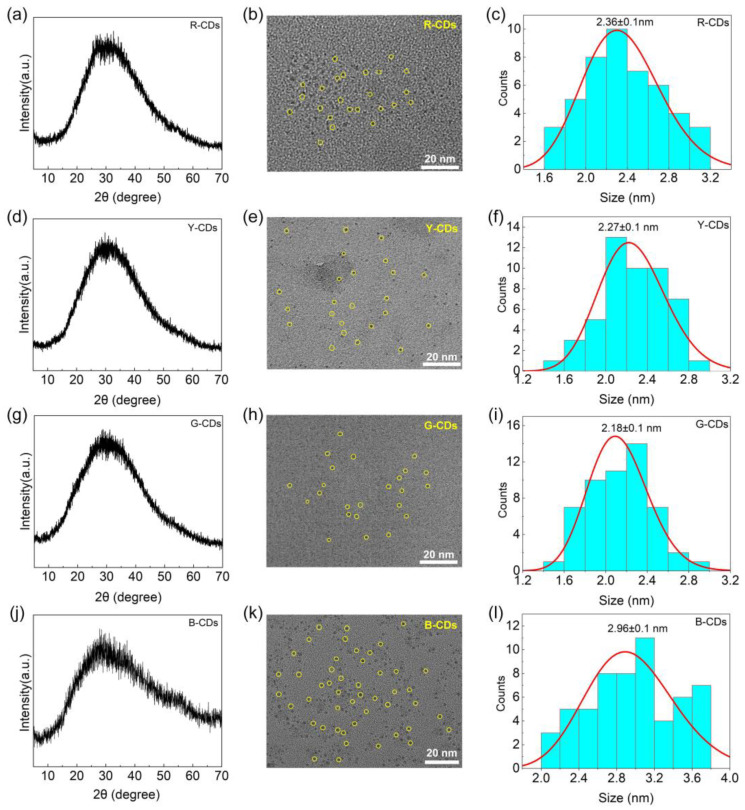
XRD patterns of R-CDs (**a**), Y-CDs (**d**), G-CDs (**g**), and B-CDs (**j**). TEM images of R-CDs (**b**), Y-CDs (**e**), G-CDs (**h**), and B-CDs (**k**); the scale bar is 20 nm. Size distributions of R-CDs (**c**), Y-CDs (**f**), G-CDs (**i**) and B-CDs (**l**).

**Figure 4 nanomaterials-12-03132-f004:**
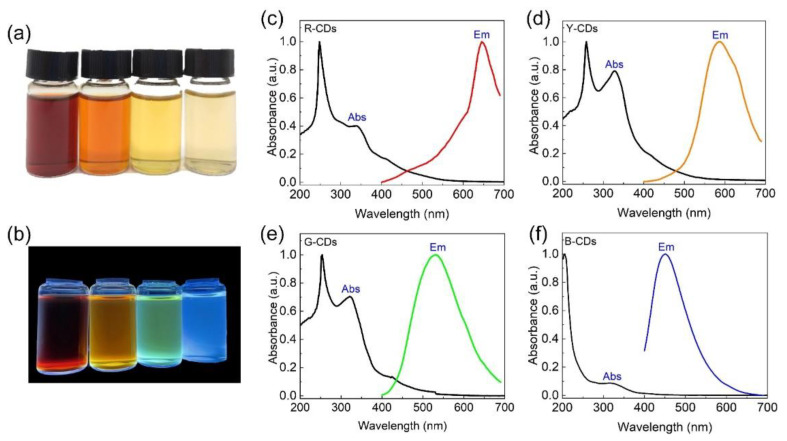
Corresponding photos of the R-CDs, Y-CDs, G-CDs, and B-CDs under sunlight (**a**) and UV light (365 nm) irradiation (**b**), respectively. (**c**) Emission and UV-adsorption spectra of R-CDs. (**d**) Emission and UV-adsorption spectra of Y-CDs. (**e**) PL emissions and UV-adsorption spectra of G-CDs. (**f**) Emission and UV-adsorption spectra of B-CDs.

**Figure 5 nanomaterials-12-03132-f005:**
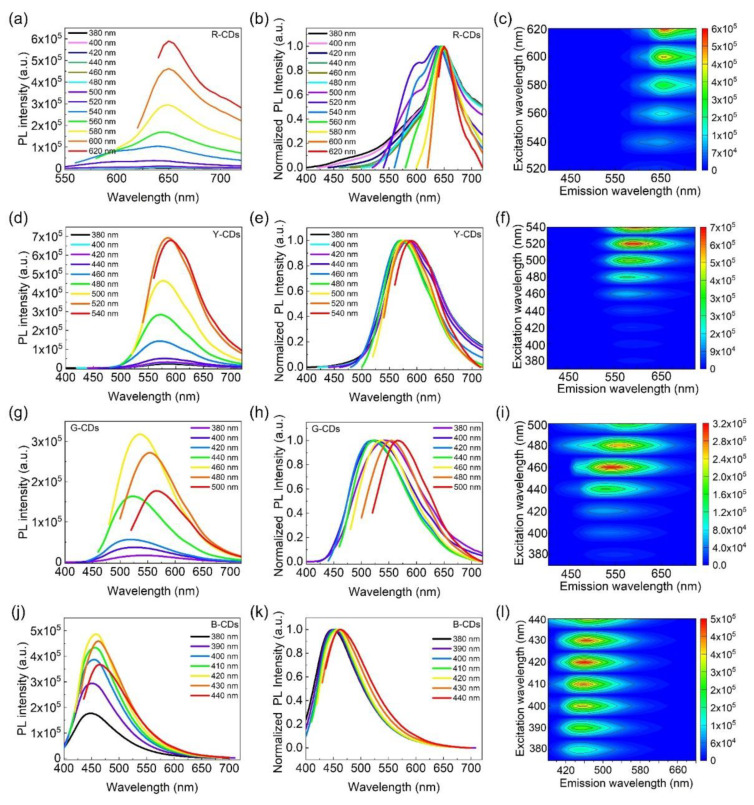
PL emission spectra of the R-CDs (**a**), Y-CDs (**d**), G-CDs (**g**), and B-CDs (**j**) under different excitation wavelengths. Normalized PL emission spectra of R-CDs (**b**), Y-CDs (**e**), G-CDs (**h**), and B-CDs (**k**) under different excitation wavelengths. Contour of the excitation spectra and emission spectra of R-CDs (**c**), Y-CDs (**f**), G-CDs (**i**), and B-CDs (**l**).

**Figure 6 nanomaterials-12-03132-f006:**
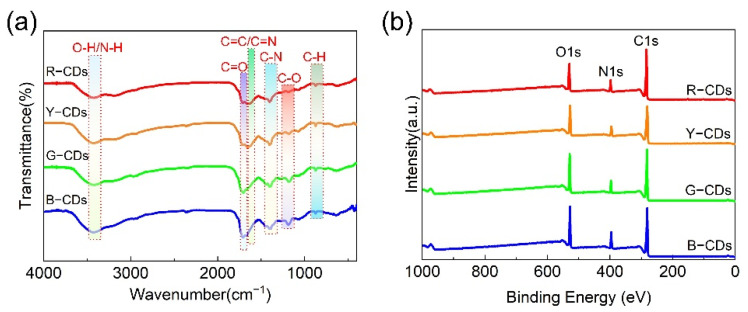
(**a**) FTIR spectra and (**b**) full-scan XPS spectra of the R-CDs, Y-CDs, G-CDs, and B-CDs.

**Figure 7 nanomaterials-12-03132-f007:**
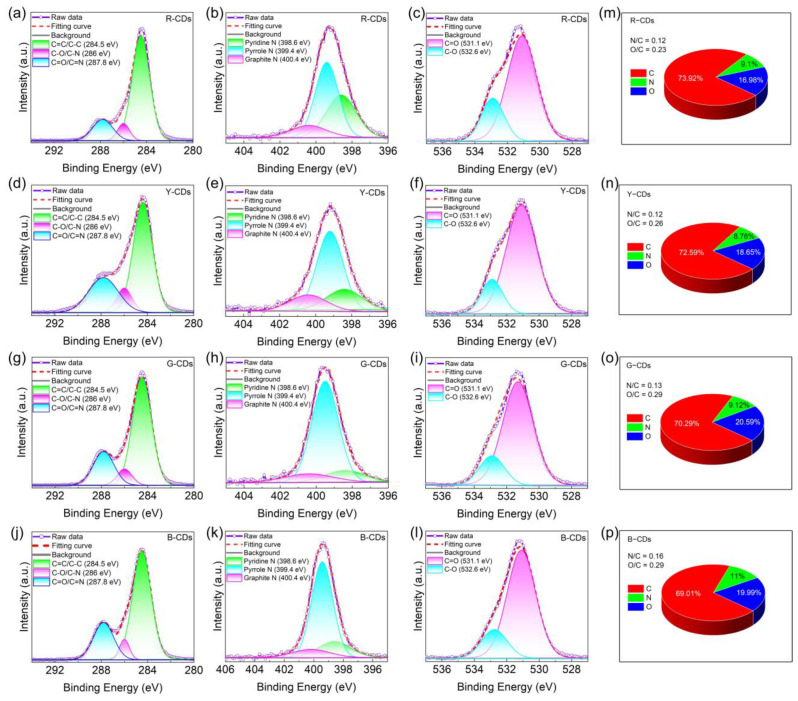
The high-resolution XPS spectra of the R-CDs (**a**–**c**), Y-CDs (**d**–**f**), G-CDs (**g**–**i**), and B-CDs (**j**–**l**) for C1s, N1s, and O1s. The quantitative analysis of XPS for the R-CDs (**m**), Y-CDs (**n**), G-CDs (**o**), and B-CDs (**p**).

**Figure 8 nanomaterials-12-03132-f008:**
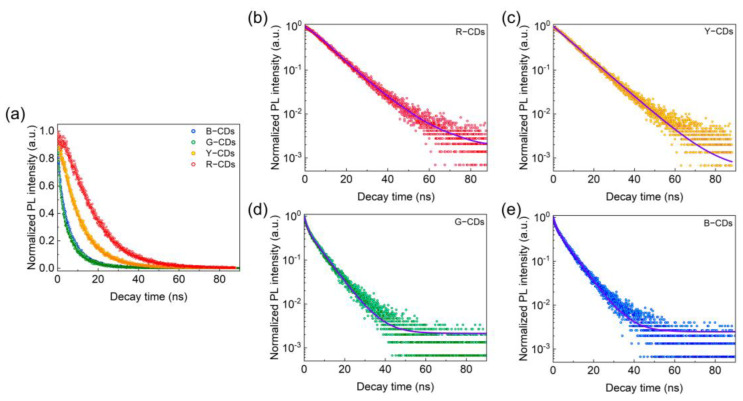
(**a**) The time-resolved PL decay curve. PL decay fitting curves of the R-CDs (**b**), Y-CDs (**c**), G-CDs (**d**), and B-CDs (**e**).

**Figure 9 nanomaterials-12-03132-f009:**
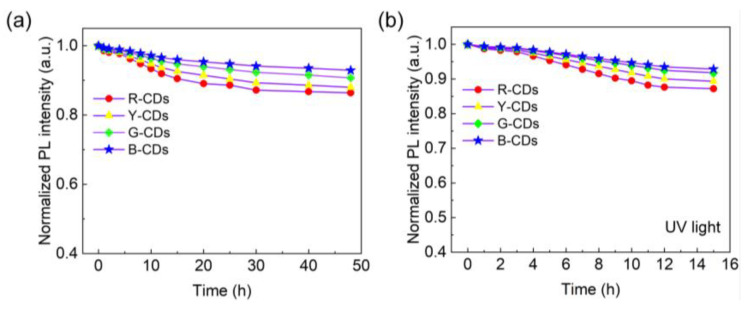
(**a**) Thermal stability of the R-CDs, Y-CDs, G-CDs, and B-CDs at 80 °C. (**b**) Photostability of the R-CDs, Y-CDs, G-CDs, and B-CDs under UV lamp irradiation.

**Figure 10 nanomaterials-12-03132-f010:**
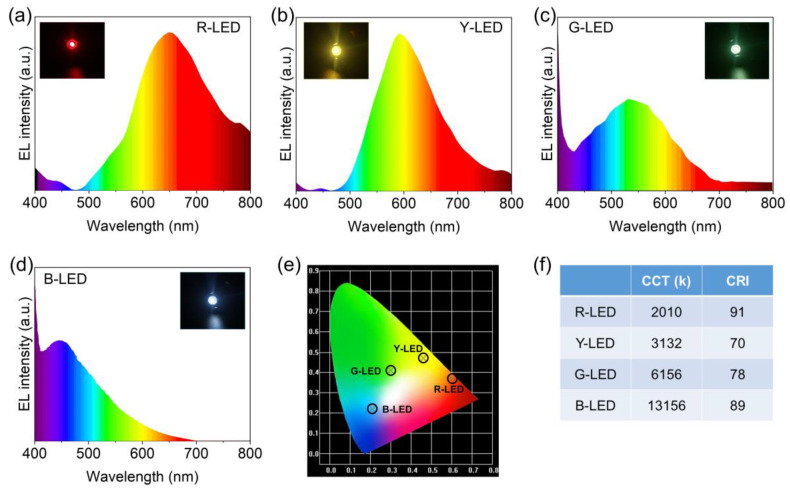
The electroluminescence (EL) patterns of the CD-based LEDs device are: (**a**) R-CD-based LED (R-LED), (**b**) Y-CD-based LED (Y-LED), (**c**) G-CD-based LED (G-LED), and (**d**) B-CD-based LED (B-LED). (**e**) CIE chromaticity coordinates of the R-LED, Y-LED, G-LED, and B-LED. (**f**) Performance of the R-LED, Y-LED, G-LED, and B-LED.

**Table 1 nanomaterials-12-03132-t001:** Fitted lifetimes of the R-CDs, Y-CDs, G-CDs, and B-CDs.

Samples	A_1_	τ_1_ (ns)	A_2_	τ_2_ (ns)	τ_avg_. (ns)
R-CDs	0.51	14.56	0.49	21.38	18.55
Y-CDs	0.50	10.76	0.50	10.86	10.81
G-CDs	0.42	1.24	0.58	6.88	6.28
B-CDs	0.33	1.19	0.67	6.72	6.27

## Data Availability

The data presented in this study are available on request from the corresponding author.

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
