# Peer review of "Multicolor Luminescent Carbon Dots: Tunable Photoluminescence, Excellent Stability, and Their Application in Light-Emitting Diodes"

_nanomaterials, 2022, doi:10.3390/nano12183132_

Round 1

Reviewer 1 Report

This is a good work and can be accepted as it is

Author Response

Thank you for your professional comments.

Reviewer 2 Report

The results are interesting and I would recommend the publication, although I would recommend to re-writte some parts in order to clarify the manuscript.

The synthesis procedure described in section 2.2 is confusing, although Figure 1 clarify in part. I would recommend to include a table with the concentration and reagents molar/ratio used for each sample.

The results and discussion section is difficult to follow and it is not clearly presented. I would recommend separating in two. First, in results section, all the results can be commented and then interpreted in the discussion section. Actually, the reader starts to read with the first paragraph (142-149) in which the conclusions are presented. Figure 2 presents a very interesting results, but before commenting those conclusions, the authors should expose which is the effect of H2O/DMF ratio, in terms of the interactions that takes place during the synthesis. The characterization  (specially the particle size) of all those samples (from 0D to 50D) should be shown (at least in the additional information section) since it is relevant to understand those results.

The nomenclarure is confusing. Sometimes it refers to the H2O/DMF ratio and sometimes the letters R, Y G B are used to refer to the colour. Since the H2O/DMF ratio is a key parameter, the paper has to clarify how it affects to the diameter size and chemical composition of the surface, in order to explain the main conclusion. At the present state of the work this point is not pretty clear.

I would say that FTIR results are not relevant, may be they can be just added at the additional information to clarity the present paper.

Author Response

Reviewer’s comments

The results are interesting and I would recommend the publication, although I would recommend to re-write some parts in order to clarify the manuscript.

Comment 1#

The synthesis procedure described in section 2.2 is confusing, although Figure 1 clarify in part. I would recommend to include a table with the concentration and reagents molar/ratio used for each sample.

Author reply

We have revised the synthesis procedure in section 2.2 and added a table with the reagents and solvents used for each sample in Supplementary Materials.

The revised sentences in section 2.2 are: “Multicolor luminescent CDs were synthesized using H2O and DMF as solvents and ammonium citrate as a precursor. Typically, red-emission CDs (R-CDs) are prepared using 3.9 g of ammonium citrate dissolved in 5 mL of H2O and 45 mL of DMF. These solutions were magnetically stirred until well-mixed. A light yellow solution was obtained by transferring the mixture to a Teflon-coated stainless steel autoclave and heating it for 8 hours at 200 °C. Then, the products were purified three times with PE to remove impurities, and they were dialyzed (1000 Da of cut-off molecular weight) for 24 hours to remove unreacted precursors. Finally, the bright emissive R-CDs solution was achieved. Other yellow-emitting CDs (Y-CDs), green-emitting CDs (G-CDs), and blue-emitting CDs (B-CDs) were obtained by only varying the volume ratio of H2O and DMF in the mixture. Other experimental procedures were carried out as described above, as shown in Table S1. In detail, 20 mL of H2O and 30 mL of DMF (20:30) were used to prepare Y-CDs, 30 mL of water and 20 mL of DMF (30:20) were used for G-CDs, and 50 mL of water (50:0) was used for Y-CDs.”

Table S1. The reagents and solvents were used to prepare each sample.

Samples

Ammonium citrate (g)

H2O

(mL)

DMF

(mL)

Time

(Hours)

Temperature

(°C)

Typical

CDs

0D

3.9

50

0

8

200

B-CDs

5D

3.9

45

5

8

200

10D

3.9

40

10

8

200

15D

3.9

35

15

8

200

20D

3.9

30

20

8

200

G-CDs

25D

3.9

25

25

8

200

30D

3.9

20

30

8

200

Y-CDs

35D

3.9

15

35

8

200

40D

3.9

10

40

8

200

45D

3.9

5

45

8

200

R-CDs

50D

3.9

0

50

8

200

 Comment 2#

The results and discussion section is difficult to follow and it is not clearly presented. I would recommend separating in two. First, in results section, all the results can be commented and then interpreted in the discussion section. Actually, the reader starts to read with the first paragraph (142-149) in which the conclusions are presented. Figure 2 presents a very interesting results, but before commenting those conclusions, the authors should expose which is the effect of H2O/DMF ratio, in terms of the interactions that takes place during the synthesis. The characterization (specially the particle size) of all those samples (from 0D to 50D) should be shown (at least in the additional information section) since it is relevant to understand those results.

Author reply

We have improved the descriptions in the Results and Discussion sections to enhance the readability of the paper.

To demonstrate the feasibility and convenience of the solvent engineering strategy, we only changed the solvent volume ratio of H2O/DMF. As shown in Figure 2, the PL emission wavelengths of the as-prepared samples can cover the visible spectrum ranging from 451 to 654 nm. Furthermore, to investigate the effect of the H2O/DMF volume ratio on optical properties, morphological features, and multicolor light-emitting mechanism of as-prepared samples, we focused on four typical CDs: R-CDs, Y-CDs, G-CDs, and B-CDs, and XRD, SEM, TEM, Abs/PL/PLE, XPS, and PL decay lifetime characterization were conducted.

Comment 3#

The nomenclature is confusing. Sometimes it refers to the H2O/DMF ratio and sometimes the letters R, Y G B are used to refer to the colour. Since the H2O/DMF ratio is a key parameter, the paper has to clarify how it affects to the diameter size and chemical composition of the surface, in order to explain the main conclusion. At the present state of the work this point is not pretty clear.

Author reply

    To make it clear, we have revised the nomenclature and added a table with the reagents and solvents used for each sample in Supplementary Materials, as presented in Table S1.

Table S1. The reagents and solvents were used to prepare each sample.

Samples

Ammonium citrate (g)

H2O

(mL)

DMF

(mL)

Time

(Hours)

Temperature

(°C)

Typical

CDs

0D

3.9

50

0

8

200

B-CDs

5D

3.9

45

5

8

200

10D

3.9

40

10

8

200

15D

3.9

35

15

8

200

20D

3.9

30

20

8

200

G-CDs

25D

3.9

25

25

8

200

30D

3.9

20

30

8

200

Y-CDs

35D

3.9

15

35

8

200

40D

3.9

10

40

8

200

45D

3.9

5

45

8

200

R-CDs

50D

3.9

0

50

8

200

 To investigate the effect of the H2O/DMF volume ratio on optical properties, morphological features, and multicolor light-emitting mechanisms of as-prepared samples, we focused on four typical CDs: R-CDs, Y-CDs, G-CDs, and B-CDs, and XRD, SEM, TEM, Abs/PL/PLE, XPS, and PL decay lifetime characterization were conducted.

By simply varying the volume ratios of the solvents, four CDs with distinct light-emitting wavelengths were created in this study. As shown in Figure 2, the PL emission wavelengths of the CDs range from 451 to 654 nm when the volume ratios of H2O/DMF fluctuate from 50/0 to 0/50, demonstrating that solvent characteristics result in emission wavelength redshift. Previous research has shown that aprotic solvents such as DMF are more likely to provide CDs precursors with a higher degree of dehydration and carbonation than protic solvents such as H2O, resulting in different sp2 carbon domains and thus a different spectrum ranging from blue to red regions [5], which is consistent with XPS characterization. In addition, solvent solubility and boiling point have a considerable impact on the chemical structure and morphology of the CDs, resulting in varied emission wavelengths. As we know, the precursor ammonium citrate is highly soluble in water but less soluble in DMF at room temperature, which can be attributed to DMF's methyl groups possessing steric barriers that decrease solvent-solvent interaction [42]. As a result, the solubility of ammonium citrate in solvent steadily decreases as more DMF is employed, and the emissions of the produced CDs shift toward a long wavelength. Furthermore, the boiling temperatures of DMF and H2O are 153 °C and 100 °C, respectively. The lower boiling point provides higher reaction pressure in the autoclave under the same reaction conditions, which greatly simplifies the doping of N and O inside produced CDs. Thus, the nitrogen and oxygen contents should gradually rise from R-CDs to B-CDs. Therefore, the polarity, solubility, and boiling point of the solvent all have an impact on the shift of CDs emission.

Comment 4#

I would say that FTIR results are not relevant, may be they can be just added at the additional information to clarity the present paper.

Author reply

    To further explicate the chemical structure of as-obtained samples, FTIR and XPS were employed to identify the chemical bonding and chemical composition of R-CDs, Y-CDs, G-CDs, and B-CDs. The FTIR and XPS results indicate that the chemical structure and degree of carbonization are related to the shifted absorption and emission bands. Therefore, the FTIR and XPS are indispensable for the characterization of CDs.

Reviewer 3 Report

The authors of this article have reported the synthesis of highly emissive luminescent quantum-sized carbon dots, exhibiting remarkably tunable and stable fluorescence emissions, from red to blue, and their characterization by X-ray powder diffraction analysis, scanning electron microscopy, transmission electron microscopy, UV-Vis absorption spectroscopy, Fourier transform infrared spectroscopy, photoluminescence spectroscopy, photoluminescence excitation spectroscopy, X-ray photoelectron spectroscopy, and photoluminescence decay lifetime. My opinion is that the topic is interesting and the manuscript falls within the scope of the Journal. The document is complete and well structured. The description of the experimental procedure is detailed and the results are critically analyzed and commented on.

I think the manuscript can be accepted in this form and it needs no revision.

Author Response

Reviewer’s comments

The authors of this article have reported the synthesis of highly emissive luminescent quantum-sized carbon dots, exhibiting remarkably tunable and stable fluorescence emissions, from red to blue, and their characterization by X-ray powder diffraction analysis, scanning electron microscopy, transmission electron microscopy, UV-Vis absorption spectroscopy, Fourier transform infrared spectroscopy, photoluminescence spectroscopy, photoluminescence excitation spectroscopy, X-ray photoelectron spectroscopy, and photoluminescence decay lifetime. My opinion is that the topic is interesting and the manuscript falls within the scope of the Journal. The document is complete and well structured. The description of the experimental procedure is detailed and the results are critically analyzed and commented on.

I think the manuscript can be accepted in this form and it needs no revision.

Author reply

Thank you for your professional comments.

Reviewer 4 Report

The manuscript “Multicolor luminescent carbon dots: Tunable photoluminescence, excellent stability, and their application in light-emitting diodes" describes an interesting technology for a robust synthesis of carbon dots with tailored spectral range. I believe both the discussion and results to be of interest to the community of the Nanomaterials. Nevertheless, a few minor issues are to be addressed to remove ambiguity for the broader audience:

- Figure 2: Why are the spectra shown only partially (e.g. “0D” and “5D” only from 400 nm up to 525 nm)? Why not to show the whole visible region proving no other features?

- Would not be it more relevant to use the volume fraction of DMF or water instead of complex “0D-50D” which constantly refers to the total volume of solvents?

- XRD discussion should be extended. I visited ref # 30 and failed to find what (110) plane the authors refer to. Please provide what structure or exact card number of an XRD database was employed to ascribe the obtained reflex on the XRD pattern. Were there any changes in the FWHM of observed reflex (the Scherrer equation)?

- Figure 3, TEM. Please indicate with arrows or dashed borders the points of interest as the images look extremely homogeneous and have low contrast.

- Figure 3, size distributions. I encourage authors to provide both the total number of dots measured as well as standard deviation values to prove such a high accuracy (+-0.01 nm!)

- Seems like lines 155-168 and 173-185 are duplicating each other.

- Experimental: please describe in detail how the film preparation for FTIR and XPS was performed

- Figure 7, XPS (m,n,o,p). Please state unambiguously whether the shown values correspond to atomic or weight fractions. Does the XPS provide such an accuracy to provide 0.01% certaintity of values?

- Why the presence of two types of emitting domains (Figure 8 and corresponding discussion) does not manifest itself on PL spectra? Are those two types of domains the same as observed on two plasmons on UV-vis absorption spectra (Fig. 4)?

- I do recommend authors check the number of significant digits (significant figures) as sometimes the accuracy is unprecedentedly high (e.g. Table 1).

- Would you be so kind to provide a photo of PDMS-based LEDs?

Author Response

Reviewer’s comments

The manuscript “Multicolor luminescent carbon dots: Tunable photoluminescence, excellent stability, and their application in light-emitting diodes" describes an interesting technology for a robust synthesis of carbon dots with tailored spectral range. I believe both the discussion and results to be of interest to the community of the Nanomaterials. Nevertheless, a few minor issues are to be addressed to remove ambiguity for the broader audience:

Comment 1#

Figure 2: Why are the spectra shown only partially (e.g. “0D” and “5D” only from 400 nm up to 525 nm)? Why not to show the whole visible region proving no other features?

Author reply

We have revised Figure 2 as follows.

Figure 2. PL spectra of the as-prepared samples that were synthesized by changing the solvent volume ratio between H2O and DMF.

Comment 2#

Would not be it more relevant to use the volume fraction of DMF or water instead of complex “0D-50D” which constantly refers to the total volume of solvents?

Author reply

    Multicolor luminescent CDs were synthesized using H2O and DMF as solvents and ammonium citrate as a precursor. The total volume of the solvents was kept constant (50 mL). As shown in Figure 2, by only changing the solvent volume ratio of H2O/DMF, the PL emission wavelengths of the as-prepared samples can cover the visible spectrum ranging from 451 to 654 nm, which fully demonstrates the feasibility and convenience of the solvent engineering strategy.

To make it clear, we have revised the nomenclature and added a table with the reagents and solvents used for each sample in Supplementary Materials, as presented in Table S1.

Table S1. The reagents and solvents were used to prepare each sample.

Samples

Ammonium citrate (g)

H2O

(mL)

DMF

(mL)

Time

(Hours)

Temperature

(°C)

Typical

CDs

0D

3.9

50

0

8

200

B-CDs

5D

3.9

45

5

8

200

10D

3.9

40

10

8

200

15D

3.9

35

15

8

200

20D

3.9

30

20

8

200

G-CDs

25D

3.9

25

25

8

200

30D

3.9

20

30

8

200

Y-CDs

35D

3.9

15

35

8

200

40D

3.9

10

40

8

200

45D

3.9

5

45

8

200

R-CDs

50D

3.9

0

50

8

200

Comment 3#

XRD discussion should be extended. I visited ref # 30 and failed to find what (110) plane the authors refer to. Please provide what structure or exact card number of an XRD database was employed to ascribe the obtained reflex on the XRD pattern. Were there any changes in the FWHM of observed reflex (the Scherrer equation)?

Author reply

The XRD patterns of R-CDs, Y-CDs, G-CDs, and B-CDs exhibit a typical carbon structure feature with significant diffraction peaks, as shown in Figures 3a, 3d, 3g, and 3j. These four CDs demonstrate a single broad diffraction peak at 2θ =29.4°, 29.6°, 29.6°, and 29.2°, respectively, which corresponds to the (002) hkl plane of the graphite carbon structure (JCPDS PDF # 75-2078) and proves the formation of a very tiny carbogenic core in CDs [30-32]. According to the Debye-Scherrer equation, the XRD pattern peaks are broadened when crystal size is decreased:

where β is the width of the observed diffraction peak at its half maximum intensity (FWHM), K is the shape factor, which takes a value of about 0.9, θ is Bragg's angle, D is the crystallite size, and λ is the X-ray wavelength (Cu-Kα radiation, equals to 0.15444 nm).

Furthermore, TEM was used to examine the morphology and size of four CDs. Figures 3b, 3e, 3h, and 3k show that the four CDs are uniform and well-dispersed. Meanwhile, four CDs' size distribution histograms (Figures 3c, 3f, 3i, and 3l) show a narrow size distribution, with average diameters of 2.36±0.1, 2.27±0.1, 2.18±0.1, and 2.96±0.1 nm, respectively. According to the TEM data, the average diameter of the corresponding CDs decreased and then increased from R-CDs to B-CDs, and some large crystals appeared. Notably, when compared to R-CDs, Y-CDs, and G-CDs, the obtained B-CDs are easy to form "clusters" due to their high surface energy and agglomeration tendency, which can explain why the particle size of B-CDs calculated by the Debye-Scherrer equation differs from that of TEM.

Comment 4#

Figure 3, TEM. Please indicate with arrows or dashed borders the points of interest as the images look extremely homogeneous and have low contrast.

Author reply

    We have added circles around the points of interest to increase the contrast with the background, as shown in Figures 3b, 3e, 3h, and 3k.

Figure 3. XRD patterns of R-CDs (a), Y-CDs (d), G-CDs (g), and B-CDs (j). TEM images of R-CDs (b), Y-CDs (e), G-CDs (h), and B-CDs (k), the scale bar is 20 nm. Size distributions of R-CDs (c), Y-CDs (f), G-CDs (i) and B-CDs (l).

Comment 5#

Figure 3, size distributions. I encourage authors to provide both the total number of dots measured as well as standard deviation values to prove such a high accuracy (±0.1 nm!)

Author reply

We have added the number of measurement points and the standard deviation of average values, as shown in Figures 3c, 3f, 3i, and 3l.

The revised sentence is “four CDs' size distribution histograms (Figures 3c, 3f, 3i, and 3l) show a narrow size distribution, with average diameters of 2.36±0.1, 2.27±0.1, 2.18±0.1, and 2.96±0.1 nm, respectively.”

Figure 3. XRD patterns of R-CDs (a), Y-CDs (d), G-CDs (g), and B-CDs (j). TEM images of R-CDs (b), Y-CDs (e), G-CDs (h), and B-CDs (k), the scale bar is 20 nm. Size distributions of R-CDs (c), Y-CDs (f), G-CDs (i) and B-CDs (l).

Comment 6#

Seems like lines 155-168 and 173-185 are duplicating each other.

Author reply

    We have corrected this mistake.

Comment 7#

Experimental: please describe in detail how the film preparation for FTIR and XPS was performed.

Author reply

FTIR test: CDs samples were dispersed in hexanol and then dropped onto potassium bromide tablets, which were dried in an infrared lamp and measured in the range of 400-4000 cm-1.

XPS test: The CDs solution samples were dropped on clean silicon wafers, dried, and then tested.

Comment 8#

Figure 7, XPS (m,n,o,p). Please state unambiguously whether the shown values correspond to atomic or weight fractions. Does the XPS provide such an accuracy to provide 0.01% certaintity of values?

Author reply

The values shown in Figures 7m-p are the atomic fractions of C, N, and O.

X-ray photoelectron spectroscopy (XPS) determinations were carried out at a Thermo Scientific machine (Thermo K-Alpha) with a mono Al-Kα excitation source (1486.6 eV) as the X-ray source. This machine can provide an accuracy of 0.01%.

Comment 9#

Why the presence of two types of emitting domains (Figure 8 and corresponding discussion) does not manifest itself on PL spectra? Are those two types of domains the same as observed on two plasmons on UV-vis absorption spectra (Fig. 4)?

Author reply

As shown in Figure 8, The existence of τ1 and τ2 implies that the CDs have two luminescence centers, coming from the π-π* transitions of the carbon core with conjugated sp2 domains and the n-π* transitions of the O-based and N-based contained functional groups, which adjust the fluorescence performance of CDs jointly.

The PL emission spectrum, normalized PL emission spectrum, and three-dimensional excitation-emission fluorescence spectrum of R-CDs, Y-CDs, G-CDs, and B-CDs under different excitation wavelengths are presented in Figure 5 to investigate the relationships between the excitation and emission properties of CDs. As the excitation wavelengths increase, the maximum emission wavelengths of R-CDs and B-CDs show no significant changes, indicating relatively stable PL emission at 650 nm and 452 nm, respectively (Figures 5a-c and 5j-l). This excitation-independent feature is most likely caused by eigenstate emission, which is closely related to the carbon core and is more similar to traditional inorganic phosphors than previously reported CDs[34]. However, as the excitation wavelengths increase, the maximum emission wavelengths of the as-prepared Y-CDs and G-CDs gradually redshift (Figures 5d-f and 5g-i), demonstrating excitation-dependent PL features, which is similar to most CDs in previous works [35,36]. The excitation-dependent PL behaviors are possibly related to the carbon bonds and surface functional groups [37,38]. These findings are consistent with UV-Vis absorption characteristics.

Comment 10#

I do recommend authors check the number of significant digits (significant figures) as sometimes the accuracy is unprecedentedly high (e.g. Table 1).

Author reply

    We have revised the number of significant digits.

Table 1. Fitted lifetimes of the R-CDs, Y-CDs, G-CDs, and B-CDs.

Samples

A1

τ1 (ns)

A2

τ2 (ns)

τavg. (ns)

R-CDs

0.51

14.56

0.49

21.38

18.55

Y-CDs

0.50

10.76

0.50

10.86

10.81

G-CDs

0.42

1.24

0.58

6.88

6.28

B-CDs

0.33

1.19

0.67

6.72

6.27

Comment 11#

Would you be so kind to provide a photo of PDMS-based LEDs?

Author reply

    2.3. Fabrication of CDs-based LED devices

We created CDs-based LEDs in a similar way to our previously published work [27]. Using R-CDs LED as an example, R-CDs was combined with PDMS in a weight ratio of 3:8 for 12 minutes using a vacuum homogenizer at 1360 rpm and 0.2 MPa vacuum. To form a phosphor layer, the mixture was dropped on the surface of the UV LED chip and solidified at 120 °C for two hours under ambient conditions. To prevent contamination and damage, a hemispherical lens was used for encapsulation.

We provide two images of PDMS-based LEDs.

Fig. 1 Schematic diagram of UV LED structure without PDMS and current (left), physical view of LED device testing after packaging (right).

Round 2

Reviewer 2 Report

The authors have addressed all my comments and suggestions.